# On-Chip E_00_–E_20_ Mode Converter Based on Multi-Mode Interferometer

**DOI:** 10.3390/mi14051073

**Published:** 2023-05-18

**Authors:** Yuan Zhang, Yuexin Yin, Yingzhi Ding, Shengyuan Zhang, Xiaoqiang Sun, Daming Zhang, Ye Li

**Affiliations:** 1School of Physics, Changchun University of Science and Technology, Changchun 130022, China; yuanz@mails.cust.edu.cn (Y.Z.); zhangsy@mails.cust.edu.cn (S.Z.); 2State Key Laboratory of Integrated Optoelectronics, College of Electronic Science and Engineering, Jilin University, Changchun 130012, China; yxyin20@mails.jlu.edu.cn (Y.Y.); dingyz22@mails.jlu.edu.cn (Y.D.); sunxq@jlu.edu.cn (X.S.); zhangdm@jlu.edu.cn (D.Z.)

**Keywords:** integrated optic, mode converter, multimode interference, integrated optics devices

## Abstract

Mode converters is a key component in mode-division multiplexing (MDM) systems, which plays a key role in signal processing and multi-mode conversion. In this paper, we propose an MMI-based mode converter on 2%-Δ silica PLC platform. The converter transfers E_00_ mode to E_20_ mode with high fabrication tolerance and large bandwidth. The experimental results show that the conversion efficiency can exceed −1.741 dB with the wavelength range of 1500 nm to 1600 nm. The measured conversion efficiency of the mode converter can reach −0.614 dB at 1550 nm. Moreover, the degradation of conversion efficiency is less than 0.713 dB under the deviation of multimode waveguide length and phase shifter width at 1550 nm. The proposed broadband mode converter with high fabrication tolerance is promising for on-chip optical network and commercial applications.

## 1. Introduction

With the development of new generation communication technology and large-capacity signal processing technology, the demand for high-speed data transmission and large-capacity data processing is increasing. Various multiplexing methods have been proposed for this purpose, such as wavelength division multiplexing (WDM), space division multiplexing (SDM) and polarization division multiplexing (PDM) technologies [1,2,3,4,5,6,7,8]. Mode division multiplexing (MDM) has been considered as a promising technology [9,10] that can significantly improve the capacity of optical communication, on-chip interconnect, and computation [11,12]. Since the orthogonal eigenmodes of the same wavelength propagate in a single channel without inter-channel crosstalk, cooperation with WDM technology, MDM technology increases overall transmission capacity dramatically [13,14]. In MDM system, mode converter with broad bandwidth, low insertion loss, and high mode conversion efficiency is the key components between fundamental mode and higher-order modes at optical nodes [15,16,17,18,19,20]. Mode converters have been demonstrated on different structures, such as asymmetrical directional couplers (ADCs) [21], Y-branch [22], multimode interferometers (MMIs) [23,24]. Although ADCs convertors show low loss and high fabrication tolerant, they suffer from high wavelength dependence. Y-branch converters contribute high crosstalk between modes. Nowadays, slotted waveguide and etched metasurface waveguide converter show ultra-compact footprint and high efficiency of mode conversion [17,25,26,27,28]. However, for commercial usage, large fabrication tolerances, in the micrometer range, are necessary. Compared with other kinds of mode conversion techniques, the mode converters based on MMIs shows broad bandwidth and large fabrication tolerance.

In this paper, we demonstrate an MMI-based mode converter on silica based planar lightwave circuits (PLC) platform with 2% refractive index difference(Δ). The light of E_00_ mode will be converted into E_20_ after propagating through the device. We optimize the mode converter through beam propagation method (BPM). The simulation result shows a conversion efficiency of −0.083 dB at 1550 nm. After fabrication, the converter shows high mode conversion efficiency of −0.614 dB at 1550 nm. From 1500 nm to 1600 nm, the conversion efficiency exceeds −1.741 dB. Moreover, the degradation of conversion efficiency is less than 0.713 dB under the deviation of multimode waveguide length and phase shifter width at 1550 nm. Compared with other mode conversion techniques, our designed MMI-based mode converter reduces the device size while ensuring high conversion efficiency, which has the advantages of broad bandwidth and large fabrication tolerance. The mode converter is suitable for large scale and complex on-chip optical networks. The high fabrication tolerance contributes the devices are promising for commercial application.

## 2. Design and Simulation

Several material platforms have been investigated for integrated photonics circuits, including silicon-on-insulator (SOI), silicon nitride (SiN), polymer-based PLC and silica-based PLC. SOI and SiN waveguide devices are compact and promising for co-package optics. However, they suffer from low fabrication tolerant and high coupling loss with fiber. Polymer based PLC devices are attractive for thermo-optical switch and high-speed modulators. The stability of polymers is a major limitation to their commercial usage. Compared with other platforms, the silica-based PLC have advantages of low loss, good match of the optical mode field with an optical fiber, and is commercially available. However, the low thermal optical coefficient (TOC) of silica contributes to high power consumption when thermo-optical tuning is used. Therefore, the silica-based PLC platform is suitable for our passive converter.

Figure 1a shows the cross section of single mode waveguide. The thickness is 4 μm. With different germanium oxide doping, the refractive index difference between claddings and cores is 2%. Refractive indices are 1.473 and 1.444 at 1550 nm, respectively. According to the calculated relationship between effective refractive indices and waveguide width shown in Figure 1b, we choose 4 µm as the height and width to maintain single mode propagation.

The schematic of the proposed MMI-based mode converter is demonstrated in Figure 2. The converter is constructed by input waveguide, tapered waveguide, 66% mode converter MMI, phase shifter waveguide, S-bend, and output waveguide. The fundamental mode is coupled into Port 1 ($P_{\mathrm{in}}$) through a tapered waveguide. The 66% E_00_ mode is converted into E_10_ at Port 2 ($P_{out}^{1}$) after the first MMI. The light left propagates in fundamental mode through Port 3 ($P_{out}^{0}$). A phase shifter is introduced to ensure π phase difference between E_00_ mode and E_10_ mode. When these lights meet at the second MMI, the mode of light is converted to E_20_ mode at Port 4 ($P_{\mathrm{out}}$).

The first MMI coupler is a 3 × 3 MMI with outputs modified into two, one of which is for first-order mode output, another is for fundamental mode output. The length of the conventional 3 × 3 MMI is equal to the length of the 66% mode converter MMI, which is $L_{MMI}=L_{3}^{1}=3L_{c}/3=Lc$, with $L_{c}$ as the coupling length based on self-imaging property and it is given by [29]:(1)$$Lc=\frac{\pi }{\beta_{0}-\beta_{1}}=\frac{4n_{r}W_{}^{2}}{3\lambda_{0}}$$
where $\beta_{0}$ is the fundamental mode propagation constant, $\beta_{1}$ the first-order mode propagation constant, $\lambda_{0}$ is the operation wavelength, $n_{r}$ is the refractive index of the core waveguide and W is the effective width taking into account the Goos–Hänchen shifts at the ridge boundaries, whose displacement is very small, and can approximatively equal to the width of MMI ($W_{MMI}$). For a conventional 3 × 3 MMI, when E_00_ mode is input from port $P_{in}$, the powers at three output ports of E_00_ modes are equal. However, the phases of the three outputs are $\varphi_{11}=\varphi_{0}+5\pi /3$, $\varphi_{12}=\varphi_{0}+2\pi /3$ and $\varphi_{13}=\varphi_{0}+\pi$, respectively, where $\varphi_{0}$ is a constant phase. Therefore, the phase difference between the two fundamental modes at Port 2 is π. We combine these two fundamental modes to form E_10_ mode. As for of the E_20_ mode, we need a 2/3π phase shifter. With BPM calculating, the width and length of the phase shifter are $W_{PS}=3.7\;\mu m$ and $L_{PS}=200\;\mu m$, respectively.

Based on the above principle, we use BPM for the design and optimization of the device at 1550 nm. To reduce the loss caused by mode mismatch, tapered waveguides are introduced between MMI region and waveguides. Widths of tapers are transferred from $W_{1}=4\;\mu m$, $W_{2}=11.5\;\mu m$ and $W_{3}=4.5\;\mu m$ to ${W^{\prime }}_{1}=4.5\;\mu m$, ${W^{\prime }}_{2}=10.4\;\mu m$, and ${W^{\prime }}_{3}=4\;\mu m$, linearly. The gap between E_10_ mode port and E_00_ mode port is 4.85 µm. The length of these three tapered waveguides is 35 µm. There are 100-µm-length straight waveguides after E_00_ and E_10_ mode port. We use S-bend to connect Port 2 with output width. The arc radius of the S-bend is above 2 cm. The loss caused by S-bend could be neglected. The length of the straight waveguide that connects phase shifter and output waveguide is chosen to be 400 µm. The width of the output waveguide is set to be 15.9 µm. In our design, the width of the MMI (*W*_*MMI*_) is designed to be 22.25 µm, which is used to avoid coupling between E_10_ mode and E_00_ mode at ports 2 and 3. Therefore, the calculated MMI coupler length is optimized to be *L*_*MMI*_ = 730 μm. The length of the output waveguide is 800 µm. The total length of the mode converter is 2400 µm.

Figure 3a shows the field distribution of the device at 1550 nm wavelength with fundamental mode is input. Figure 3b shows the fundamental mode field distribution of the input. The mode distributions of E_10_ at port 2 and E_00_ at port 3 of the mode converter MMI are shown in Figure 3c. Figure 3d shows the mode distribution of E_20_. From Figure 3a, the E_00_ mode is successfully converted to the E_20_ mode. In the simulation, we set the E_00_ mode input power of port1 to 1 and simulated the conversion efficiency of the mode converter from 1500 nm to 1600 nm, as shown in Figure 3e. The conversion efficiency in percent (*E_p_*) and dB (*E_d_*) are calculated by Equations (2) and (3), respectively. Here, b is the propagation loss of the E_00_ mode in the dB scale [30].
(2)$$E_{p}=(1-10^{0.1b})\times 100\%$$
(3)$$E_{d}=10\mathrm{lg}(1-10^{0.1b})$$

At 1550 nm wavelength, the conversion efficiency is 96.2% (−0.083 dB). From 1500 nm to 1600 nm, simulation results show that the conversion efficiency exceeds −0.630 dB. In summary, we obtain a mode conversion from E_00_ to E_20_ at 1550 nm with a conversion efficiency of −0.083 dB.

## 3. Characterization and Discussion

The proposed mode converter was fabricated by the PLC foundry, SHIJIA, China. First, the bottom cladding of ~15 µm was thermally oxidized on silicon substrate. Second, we grew a 4-µm-thickness silica core layer above the bottom cladding by plasma enhanced chemical vapor deposition (PECVD). The waveguide was formed by etching through inductively coupled plasma etching (ICP) technique. Finally, PECVD was again used to deposit the upper cladding. Since the existing coupling test system is difficult to directly monitor the high-order mode power, a mode demultiplexer was introduced at the end of the output waveguide to convert the E_20_ mode to the E_00_ mode to indirectly characterize the higher-order mode power. The mode multiplexer/demultiplexer of E_20_ mode and a reference waveguide are fabricated simultaneously. The microscope images of the demonstrated converter are shown in Figure 4a–d.

In Figure 4a, the measured width of the MMI is 22 μm. The widths of the two output ports are measured in Figure 4b at 10 μm and 4 μm, respectively. The S bend is separated from the phase shifter by a large distance of 5 μm, which can be observed in Figure 4c. The width of the output waveguide is measured to be 16 μm in Figure 4d.

The E_00_ mode is coupled into the device by edge coupling system. The spectra of the reference straight waveguide and the mode multiplexed/demultiplexer for the E_20_ mode are shown in Figure 5a. The normalized spectral response of the E_00_ to E_20_ mode converter experimental measure data is shown in Figure 5b. The experimental results show that the conversion efficiency can exceed −1.741 dB with the wavelength range of 1500 nm to 1600 nm. The conversion efficiency of the device is −0.614 dB (87%) at 1550 nm. The maximum conversion efficiency is −0.549 dB (88%) at 1556 nm. The 1-dB bandwidth of the device is 90 nm. The captured output mode patterns at different wavelengths shown in Figure 5c–e.

We fabricate proposed device with different lengths of MMI and widths of phase shifter to perform fabrication tolerant. The key components, MMIs and phase shifters, influence the conversion efficiency of the device a lot. We study the fabrication tolerance by changing length and width of MMI and phase shifter, and observe the output power of E_20_. The deviations are 0 and ±10 µm for *L_MMI_* and 0 and ±0.2 µm for *W_PS_*. The simulation result is shown in the Figure 6. We can see that when *L_MMI_* move away from their optimal values, the conversion efficiency of E_00_–E_20_ has different degrees of degradation at wavelength 1550 nm., which means that the change of *L_MMI_* will cause an impact on the conversion efficiency of the device. At 1550 nm, the conversion efficiency varies from −0.083 dB to −0.234 dB.

Similarly, the influence of *W_PS_* on the conversion efficiency of the device is simulated. Figure 7 shows the dependence of the E_00_–E_20_ conversion efficiency on the *W_PS_*. We use the optimal value *L_MMI_* = 730 μm as the length of MMI. It can be seen that the change of *W_PS_* does have a certain impact on the conversion efficiency of the mode converter. At the wavelength of 1550 nm, the conversion efficiency of the mode converter varies from −0.083 dB to −0.155 dB. Compared with *L_MMI_*, the device is more sensitive to *W_PS_*.

We summary the spectra of all samples and separate in four groups, as shown in Figure 8a–d. It can be seen that at the wavelength of 1550 nm, the conversion efficiency degradation of the device is less than 0.713 dB when changing *L_MMI_* and *W_PS_*. However, the conversion efficiency of the converter proposed is lower than others. So there are still some cases that need to be discussed for the MMI-based mode converter we designed. First of all, during the simulation, we initially expected the conversion efficiency to be −0.083 dB (97%) at 1550 nm. However, the actual test results show that the conversion efficiency of the mode converter can only reach −0.614 dB (87%) at 1550 nm. The reason is the differences in geometry and refractive index of cores with respect to design parameters. The device has good repeatability, parameter scanning will help but only if you also have good repeatability. This problem is solved by parameter scanning while fabrication. Secondly, in Figure 8a,d, we note that when *L_MMI_* = 720 μm the normalized transmission may exceed 0, which is caused by the normalization procedure owing to the vibration of the reference waveguide.

## 4. Discussion

In order to clearly demonstrate the mode converter, Table 1 illustrates the performance comparisons between the reported E_00_–E_20_ mode converters. The compact silicon-on-insulator based device using a 1 × 4 Y-junction and 4 × 4 MMI couplers can realize 16 different input-output mode conversions in C-band [31]. The mode converter with a cascaded tapers structure exhibits balanced performances [32]. The mode converter designed in Ref. [33] with phase change material inlaid in multimode waveguide has a large bandwidth and high conversion efficiency. However, the mode converters in these reports only have simulation results and are not actually fabricated. Owing to the low fabrication tolerance of the SOI platform, the device is hard to fabricate with the designed characteristics. Moreover, the SOI device is fixed after fabrication. To obtain a flexible and reconfigurable device, mode switching is necessary for the MDM system. The mode switch based on phase change materials or van der Waals materials can be used to achieve low power consumption and higher-order reconfigurable mode conversion [33,34,35]. In ref. [36], a mode converter based on cascaded long-period waveguide grating is mentioned, which uses the UV-curable polymer material EPO as the design material. The conversion efficiency of the device can be estimated to be more than 90% at 1550 nm. The polymer-based PLC is a low-cost platform. The design method could be verified on polymer. The design method is also inspiring for silica-based PLC. After comparison, the device proposed shows high conversion efficiency, high tolerance and large bandwidth, which has a good application prospect in MDM system.

## 5. Conclusions

In this work, we proposed an MMI-based mode converter that has the capability to efficiently convert E_00_ mode to E_20_ mode. The experimental results show that the fabricated device has high tolerance and high conversion efficiency, and the efficiency of the fabricated mode converter is −0.614 dB at 1550 nm wavelength. The degradation of conversion efficiency is less than 0.713 dB within the range of fabricate deviation. This paper provides a solution for efficient conversion in MDM systems.

## Figures and Tables

**Figure 1 micromachines-14-01073-f001:**
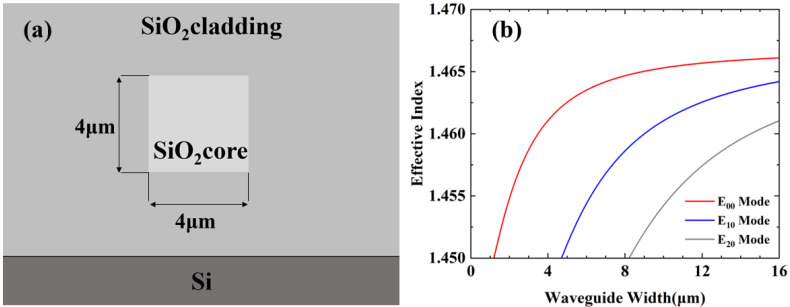
(**a**) Cross section of the silica waveguide. (**b**) Calculated effective indices of 4-μm-thickness silica waveguide with different widths.

**Figure 2 micromachines-14-01073-f002:**
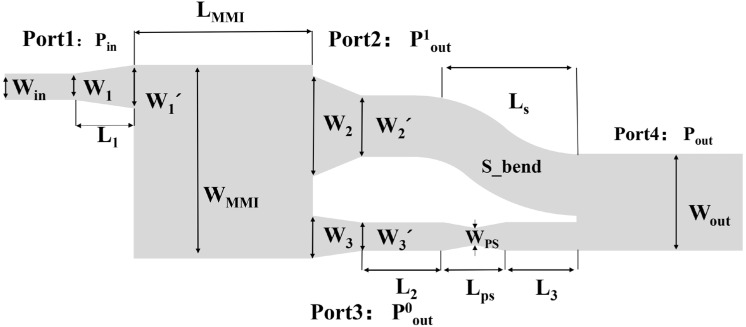
Schematic of the MMI-based mode converter.

**Figure 3 micromachines-14-01073-f003:**
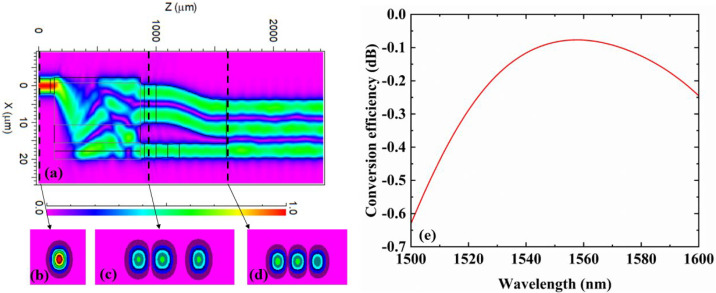
(**a**) Simulated optical field propagation distribution of the device at 1550 nm. (**b**) Mode profile of E_00_ at port 1. (**c**) Mode profile of E_10_ at port 2 and E_00_ at port3. (**d**) Mode profile of E_20_ at port 4. (**e**) Simulated conversion efficiency of the mode converter at 1500 nm to 1600 nm.

**Figure 4 micromachines-14-01073-f004:**
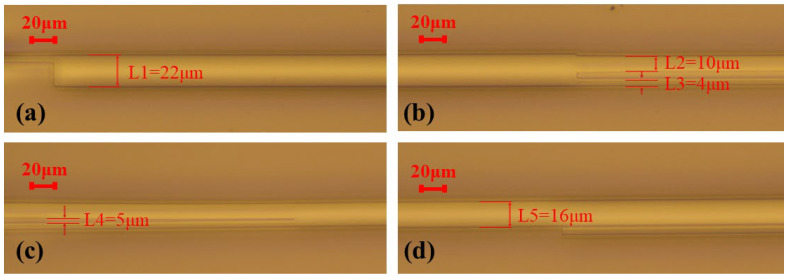
(**a**–**d**) Microscopic images of the fabricated mode converter.

**Figure 5 micromachines-14-01073-f005:**
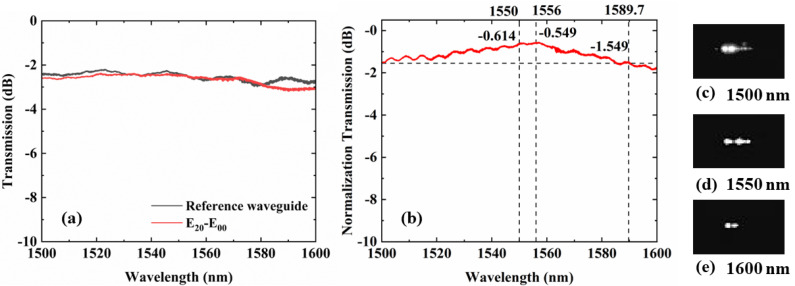
(**a**) Measured power loss of the reference waveguide with the E_20_ mode (de) multiplexer. (**b**) Conversion efficiency of the mode converter from E_00_ to E_20_ in the wavelength range from 1500 nm to 1600 nm. Output mode taken from output port at (**c**) 1500 nm, (**d**) 1550 nm and (**e**) 1600 nm.

**Figure 6 micromachines-14-01073-f006:**
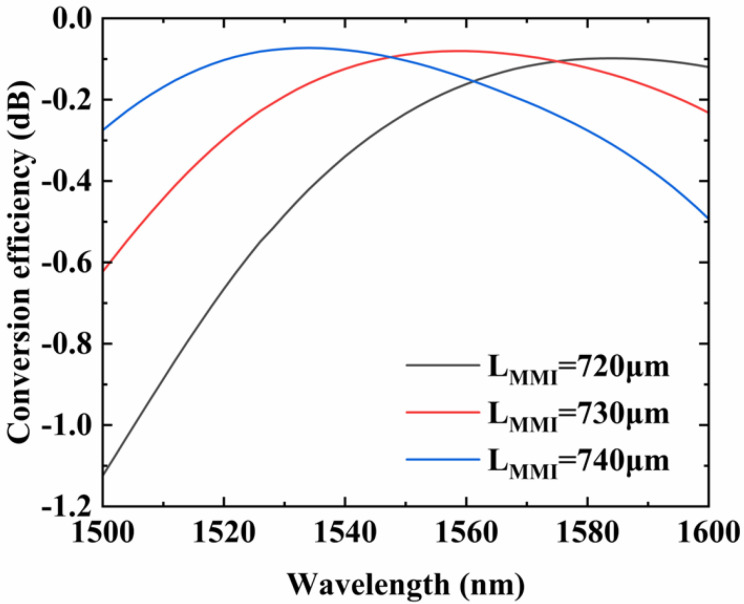
Simulated conversion efficiency of the mode converter when changing the *L_MMI_* from 1500 nm to 1600 nm.

**Figure 7 micromachines-14-01073-f007:**
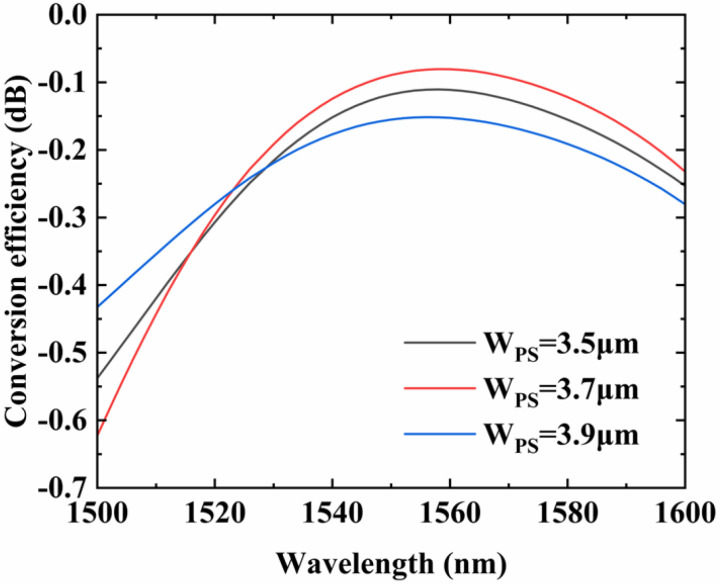
Simulated conversion efficiency of the mode converter when changing the *W_PS_* from 1500 nm to 1600 nm.

**Figure 8 micromachines-14-01073-f008:**
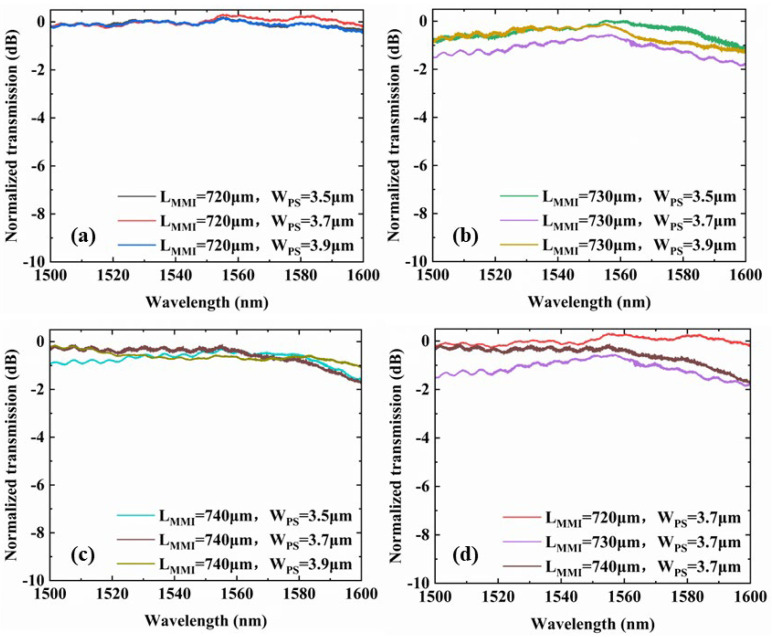
(**a**–**c**) The effect of $W_{PS}$ on the conversion efficiency of the mode converter at a given fabrication error range for wavelengths ranging from 1550 nm to 1600 nm with *L_MMI_* of 720 µm, 730 µm and 740 µm, respectively. (**d**) The effect of *L_MMI_* on the conversion efficiency of the mode converter at a given fabrication error range for wavelengths ranging from 1550 nm to 1600 nm when $W_{PS}$ is fixed to 3.7 µm.

**Table 1 micromachines-14-01073-t001:** Comparison the designed E_00_–E_20_ mode converter with reported works.

Refs.	S/E	BW (nm)	CE (%)	Structure	Materials	Footprint (µm^2^)
[29]	E	35	93	MMI+ phase shifter	SOI	1230 × 8
[31]	S	35	98	Y-junction + MMI	SOI	288 × 6.8
[32]	S	100	93	Cascaded tapers	SOI	2.5 × 6.5
[33]	S	245	99	PCM inlaid in multimode waveguide	SOI	2.3 × 1.82
[34]	E	At 1550 nm	90	Long-period gratings	Polymer PLC	/
This work	E	90	87	MMI + phase shifter	Silica PLC	2400 × 22.25

S/E, Simulation/Experiment; BW, Bandwidth; CE, Conversion efficiency; PCM, Phase change material; /, Not mentioned.

## Data Availability

Not applicable.

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
