# Peer review of "On-Chip E00–E20 Mode Converter Based on Multi-Mode Interferometer"

_micromachines, 2023, doi:10.3390/mi14051073_

Round 1

Reviewer 1 Report

The paper proposes an MMI-based mode converter that can efficiently convert E00 mode to E20 mode with high fabrication tolerance and large bandwidth. The experimental results show that the fabricated device has high tolerance and high conversion efficiency, and the efficiency of the fabricated mode converter is -0.614 dB at 1550 nm wavelength. The degradation of conversion efficiency is less than 0.713 dB within the range of fabricate deviation. The following issues need to be addressed before considering publishing this article.

1. Please check φ130+2π/3, in the ref. 25, φ110+π.

2. Line 92 to line 102, the author needs to provide a more detailed description of the device design process, including the basis for selecting structural parameters.

3. The author mentions that "From 1500 nm to 1600 nm, simulation results show that the conversion efficiency exceeds -1.741 dB." However, this information is not apparent from Figure 3b.

4. Figure 4 is redundant.

5. Some values in the data lines of Fig. 6(a) and Fig. 6(d) exceed >0 dBm. The author is requested to explain this situation. Additionally, it is recommended that the author use simulations to demonstrate the device's manufacturing tolerance and reorganize the description of Figure 6 accordingly.

6. In Table 1, the author claims that the mode conversion efficiency of the device reaches 87% and the bandwidth is 100nm. However, this information cannot be obtained from Figure 5(b), and the author needs to explain this.

7. Additionally, it is recommended to modify Table 1 to include a column indicating the device platform, such as SOI, silica-on-silicon, etc., and add another column for the device size.

Reviewer 2 Report

In the manuscript, the authors presented a compact MMI-based mode convertor at optical telecommunication wavelength with fair conversion efficiency. Theoretical analysis and experimental results are presented with good agreements.

The manuscript is organized in a logical manner with overall complete results. Nevertheless, before recommending publication, the following technical comments should be addressed by the authors.

1. One major concern is about manuscript novelty. In the introduction section, the authors properly reviewed the status and drawbacks of conventional mode convertors. However, the authors missed an important category of subwavelength meta-structures-based mode convertors (such as Ref. Nature Nanotechnology,12, 675–683, 2017. Ref: Photonics Research 8, 564-576, 2020. Ref: Light: Science & Applications 10, 235, 2021), which also feature compact size and high conversion efficiency.

The authors should benchmark and compare their proposed devices to the meta-devices mentioned above to further highlight the novelty of this manuscript.

2. Currently the mainstream of integrated photonics are based on Si or SiN, which have comparatively higher index or stronger light confinement than SiO2 for the sake of compact photonic devices.

Despite that the advantages of PLC are briefly mentioned in Section 2, the authors should further comment on their motivation of using silica as the waveguiding media, or it it could be competitive if compared to Si photonics.

3. Some figure arrangements can be optimized.

(a) For instance, some font sizes in Fig. 1 is much smaller than others. Fig. 1 and Fig. 2 could be combined into one figure with subpanels (a) and (b) to save space.

(b) The scale bar and colorbar are missing in Fig. 3a. 

4. For Fig. 4, some delicate structure details are typically difficult to clearly show out in optical microscope images. The authors can consider to supplement SEM images to the proposed device for better evaluate the device fabrication between theoretical design.

5. One of the major challenging in mode convertor and other integrated photonic devices is the lack of reconfigurability: i.e. devices cannot be effectively tuned after fabrication.

The authors should comment on this regard adding the mentioned references potential and discussions on using, for instance, phase change materials (Ref: Nature Photonics, 11, 465–476, 2017. DOI: 10.1038/NPHOTON.2017.126 ), or van der Waals materials (Ref: Nature Reviews Materials, 2023. DOI: 10.1038/s41578-023-00558-w ) to address this issue.

The manuscript is relatively well-written in good English.

Reviewer 3 Report

The author should go through the manuscript and improve the writing. There are various places te grammatical mistakes are present. For example, line No. 56, should be ‘thermo-optical tunning’, line No. 68, should be ‘is coupled into’, line No. 169, should be ‘that has the capability to’ etc. When using an abbreviation for the first time in the manuscript, the author should write the full form as well. For example, what does ‘PLC’ stand for, etc.

Round 2

Reviewer 1 Report

The author has addressed my questions sufficiently and I recommend publication.

Reviewer 3 Report

There has been a sufficient improvement in the manuscript and it can be considered for publication in this journal.